# Synergic Activity Against MCF-7 Breast Cancer Cell Growth of Nanocurcumin-Encapsulated and Cisplatin-Complexed Nanogels

**DOI:** 10.3390/molecules23123347

**Published:** 2018-12-18

**Authors:** Ngoc The Nguyen, Ngoc Nhat Thanh Nguyen, Ngo The Nhan Tran, Phung Ngan Le, Thi Bich Tram Nguyen, Ngoc Hoa Nguyen, Long Giang Bach, Vu Nguyen Doan, Ha Le Bao Tran, Van Thu Le, Ngoc Quyen Tran

**Affiliations:** 1Department of Pharmacy and Medicine, Tra Vinh University, Tra Vinh City 940000, Vietnam; nnt.the@gmail.com; 2Department of Chemistry, Graduate University of Science and Technology, Vietnam Academy of Science and Technology, 1A TL29, District 12, Ho Chi Minh City 700000, Vietnam; 3Institute of Applied Materials Science, Vietnam Academy of Science and Technology, 1A TL29, District 12, Ho Chi Minh City 700000, Vietnam; thanhnnnt0101@gmail.com (N.N.T.N.); tranngothenhan@gmail.com (N.T.N.T.); vanialai.yin@gmail.com (P.N.L.); 4Department of Natural Science, Thu Dau Mot University, Thu Dau Mot City 590000, Vietnam; ntbtram.c44@moet.edu.vn; 5German Vietnamese Technology Center-HCMC University of Food Industry, Tan Phu District, Ho Chi Minh City 700000, Vietnam; hoacntp@cntp.edu.vn; 6NTT Hi-Tech Institute, Nguyen Tat Thanh University, 300A Nguyen Tat Thanh, Ward 13, District 4, Ho Chi Minh City 700000, Vietnam; blgiang@ntt.edu.vn; 7University of Science, Vietnam National University, Ho Chi Minh City 700000, Vietnam; dnvu@hcmus.edu.vn (V.N.D.); tlbha@hcmus.edu.vn (H.L.B.T.)

**Keywords:** dual drugs, delivery, nanocarriers, cisplatin, nanocurcumin

## Abstract

Nanogel-based systems loaded with single anticancer drugs display miscellaneous effectiveness in tumor remission, gradually circumventing mutation and resistance in chemotherapy. Hence, the existence of dual-drug delivered nano-sized systems has been contemporaneous with drug development and preceded the conventional-dose chemotherapy. Among outstanding synergistic drug nanoplatforms, thermosensitive copolymer heparin-Pluronic F127 (Hep-F127) co-delivering cisplatin (CDDP) and curcumins (Cur) (Hep-F127/CDDP/Cur) has emerged as a notable candidate for temperature-responsive drug delivery. The procedure was based on the entrapment of curcumin into the hydrophobic core of bio-degradable co-polymer Hep-F127 while the hydrophilic drug CDDP subsequently conjugated to the backbone heparin to form the core-shell structure. The copolymer was characterized by Fourier transform infrared (FT-IR) spectrophotometry, Transmission Electron Microscopy (TEM), and Dynamic Light Scattering (DLS), to corroborate the successful synthesis and via HPLC along with AES-ICP to evaluate the high drug loading along with a controllable release from the nano-gels. A well-defined nano-shell with size in the 129.3 ± 3.8 nm size range could enhance higher the efficacy of the conjugated-CDDP to Hep-F127 than that of single doses. Moreover, the considerable amount of dual-drug released from thermosensitive nanogels between different conditions (pH = 7.4 and pH = 5.5) in comparison to CDDP from Hep-F127 partially indicated the significantly anti-proliferative ability of Hep-F127/CDDP/Cur to the MCF-7 cell line. Remarkably, drug testing in a xenograft model elucidates the intricate synergism of co-delivery in suppressing tumor growth, which remedies some of the problems affecting in cancer chemotherapy.

## 1. Introduction

Breast cancer has been one of the most prevalent oncological problems with a high research demand [1]. Commercial chemotherapy drugs based on NCI’s Cancer Drug Information such as abitrexate (methotrexate), abemaciclib, erlotinib, gefitinib, paclitaxel, and cisplatin dominate in the first line of treatment portfolio. Notably, cisplatin was successfully applied as a compelling agent, which accounted for nearly half of preferably platinum (II)-complex-chemotherapeutic descriptions. Notwithstanding the interaction with nucleophilic molecules in cancer cells, it could quickly reduce pharmacokinetic tolerance due to its binding with plasma proteins [2]. Moreover, several pieces of evidence suggest that breast cancer patients cope with the drug resistance problems via intrinsic and acquired mechanisms [3,4]. Nano-formulation drug delivery technologies appeared next, improving the rapid elimination of the drug from the bloodstream and were hailed as the leaders of the next generation of carriers to ferry cancer drugs to tumors [5,6]. Moreover, the assistance of biodegradable materials, particularly body-friendly polymeric agents to fabricate nano-platforms has become a norm of research [7,8]. With the expansion of nanocarriers, there has been a the shift from mono-therapy to synergetic combinations which could boost the efficacy of clinical outcomes as well as non-overlapping toxicity [9,10,11,12]. In addition to the use of chemo-compounds in chemotherapy, approaches involving phytochemicals have led to potent complementary treatments [13,14,15,16].

Of all phytocompounds, curcumin is the most popular agent in oncology-combined pharmaceutical work. Recent studies have been shown curcumin promoted the therapeutic efficacy of Nutlin-3a in the co-delivery of folate poly(lactic-co-glycolic acid (PLGA) nanoparticle to suppress p53 protein [17]. The synergetic combination of curcumin and gemcitabine highly inhibited the tumor proliferation along with alleviating the toxicity [18]. Folate lipid nanoparticles could target into MCF7 cell line with the synergism of paclitaxel and curcumin [19]. Doxorubicin (DOX)/Cur nanoparticles inducing apoptosis in hepatocellular carcinoma, etoposide/Cur delivered by lipid nanostructures in gastric oncology regimens, camptothecin/Cur encapsulated in chitosan-modified polymeric nanoparticles to treat colon cancer have subsequently proved these trigger facilitated combined agents to be much more effective [20,21,22]. 

Due to the prominence of the abovementioned highlighting properties, the synergism of curcumin and cisplatin was investigated in this article. The thermosensitive copolymeric nano-platform Hep-F127, with outstanding controllable slow CDDP release functions was discovered in our previous research [23]. Hence, in this work we examine the effects of synergistic co-delivery of Cur/CDDP by a temperature-responsively core-shell nanostructured carrier in the suppression of breast cancer as well as minimizing the multidrug resistance (Figure 1).

## 2. Results and Discussion

### 2.1. Characterizations of the Amphiphilic Hep-F127

The structure of the grafted copolymers was well-characterized by ^1^H-NMR and FT-IR as referred in our previous report [23]. In the study, the critical micelle concentrations (CMC) of the copolymer solution were verified using the fluorescence intensity ratios of pyrene excitation bands. It is well-known that Pluronic F127, consisting of hydrophilic Hep and PEO segments as well as hydrophobic PPO segments, mediate, the self-assembly of grafted copolymer in aqueous solutions. The micellization of amphiphilic Hep-F127 in aqueous solution was investigated using pyrene as a fluorescence probe, which can provide microscopic information regarding the nature of micellar aggregates in aqueous solution. It was reported that once the micelles form in aqueous solution, the hydrophobic pyrene can be encapsulated into the hydrophobic PPO microdomains of Hep-F127 causing a decrease of the intensity ratio of the first to third band (I1/I3) in the pyrene fluorescence emission spectrum. Our result showed that the CMC of Hep-F127 is 0.174 mg/mL. This CMC value is about an order of magnitude larger than the CMC value of F127, which is 0.035 mg/mL [24]. In fact, by increasing the hydrophilic portion of a surfactant-like molecule its theoretical hydrophilic-lipophilic balance (HLB) should increase, determining an increase in the hydrophilicity of the polymer that would lower its tendency towards hydrophobic assembly. Another factor that could influence a lower hydrophobic interaction of the co-polymer is the increased steric hindrance between the (bigger) polymer chains. Hence, the result provides clear evidence of the thermo-responsive behavior of the amphiphilic Hep-F127 copolymer that would be significant in fabricating drug delivery nanocarriers.

### 2.2. Characterizations of Nano-Complexes

To structurally characterize the nano-complex, the FT-IR method was applied to Hep-F127, Hep-F127/CDDP, and Hep-F127/CDDP/Cur, giving the spectra shown in Figure 2. 

In the absorbance bands of Hep-F127 copolymer, a peak at 2888 cm^−1^ was observable, which corresponded to the CH_2_/CH_3_-stretching vibration, whereas, the –COO- and SO_2_-O- scissor bands of Hep-F127 copolymer were shifted in the range from 1629 cm^−1^ to 1592 cm^−1^ and 1148 cm^−1^ to 1109 cm^−1^, respectively, due to the complexation between the platinum hydroxide moiety of CDDP-OH with carboxylate or sulfate groups of Hep-F127 after conjugation to CDDP. Some absorbance peaks of the original curcumin were visible around 1628 cm^−1^ (C=O) and 1509 cm^−1^ (C=C). The spectrs confirmed that curcumin was encapsulated and retained its structure in the Hep-F127/CDDP nanocomplex.

The morphology of the complexed Hep-F127/CDDP/Cur nanogel were evaluated via TEM images, in which size of the F127-Hep micelles around 168.4 ± 5.3 nm (Figure 3a). Apparently, at lower critical solution temperature (LCST), the thermosensitive polymer underwent a globular-coil transition of hydrophobic segments which made the Hep-F127 nanoparticles easily visible to observe. According to our previous report, the use of a co-solvent (ethanol and dichloromethane) in above method could provide curcumin particles dispersed in the amphiphilic chitosan copolymer solution which could be utilized for several biomedical applications [25,26]. In addition, a complexation of carboxylate and sulfate groups with the aquated cisplatin was also well-demonstrated in our other previous study, that made zeta potential of Hep-F127 micelles shift from negative to positive charge [23]. In the study, the zeta potential of the grafted Hep-F127 nanogels was around −14.18 ± 0.96 mV, significantly lower than that of the complexed Hep-F127/CDDP nanoparticles (around 7.39 ± 1.4 mV). In our additional experiments, zeta potential measurements of colloidal curcumin solution showed a value close to –25.9 mV. Encapsulation of high-negative-charged and hydrophobic Cur nanoparticles inside the complexed nanoparticles reduced the potential value to −4.8 ± 0.3 mV. These evidences could indicate that the complexed Hep-F127/CDDP/Cur nanogels encapsulated curcumin in the nanosized particles as seen in Figure 3b due to the hydrophobic and negative-charge properties of curcumin. 

### 2.3. Dual Drug Releasing Profiles from the Nanocarrier

Along with the effective loading yield, dual drugs leakage from the complexed Hep-F127/CDDP/Cur nanocarriers should be controlled to the maintain bioavailability of the bioactive molecules and drugs. Indeed, cumulative drug release of CDDP and Cur occurred at a faster rate under acidic (pH 5.5) conditions than at pH 7.4, as shown in Figure 4. Particularly, the cumulative releases of CDDP and Cur were achieved greater 60% and around 80% at pH = 7.4 and pH = 5.5, respectively, after 96 h. Moreover, the cumulative CDDP leakage out from the co-delivery system was 3-fold higher compared to the initial one-hour release. This indicates that the complex between the CDDP and anionic groups (in Hep-F127) is unstable and undergoes hydrolysis in acidic media [23,27]. The incomplete platinum release seen after 96 h could be due to the conjugation of the aquated cisplatin to the amine/amide groups within the polysaccharide chains. Curcumin has very low solubility so the its release profile as a control model wasn’t examined in the study. In our previous study, for CDDP solution without Hep-F127, the released CDDP reached to 90% after 3 h. Hence, this difference in release profile behaviour apparently indicates the significance of CDDP and Cur loaded the complexed nanogels in minimizing the toxicity of the dual drugs delivery system.

### 2.4. Cytotoxicity Assay

Cell inhibition assays were carried out to determine the IC_50_ concentrations for each formulation (CDDP, Hep-F127/CDDP, Hep-F127-Cur, Hep-F127/Cur/CDDP) towards the human breast cancer cell line MCF-7. The cytotoxicity results in Figure 5 indicate that Cur-encapsulated nanogel, Hep-F127/CDDP and dual-drug delivered nanogel exhibited obvious and effective anti-proliferative activity on the cancer cells, in which 100 ppm of Hep-F127/Cur reduced cell growth by 63.10 ± 1.91%. cell Inhibition of 88.57 ± 1.38% and, notably, 95.32 ± 2.57%, was observed after treatment with Hep-F127/CDDP and Hep-F127/Cur/CDDP, respectively. CDDP was tremendously toxic, with 92.15 ± 0.85% of the cell growth being inhibited under screening concentration (100 ppm, data not included in Figure 5). while a cytotoxicity assay of the hep-F127 nanocarrier showed that the platform did not show inhibitory effect (−2.51 ± 2.44%) at the screening concentration of 100 ppm. 

Activity against MCF-7 breast cancer cell growth of the nanocurcumin-encapsulated and cisplatin-complexed nanogel was significantly increased when nanocurcumin and CDDP-OH were co-formulated at above 20 ppm of each. The results strongly confirmed that the combination of the dual components in the drug delivery system led to a synergistic activity against MCF-7 breast cancer cell growth. CDDP has been used as one of the most efficacy-spectrum chemotherapeutics by crosslinking the DNA and inhibiting DNA synthesis, resulting in cell apoptosis, however the exclusively long-term administration enhances drug resistance. Curcumin is an antioxidant possessing both anti-inflammatory and anti-tumor activities. Some clinical evidence has indicated that the combination of curcumin with anticancer drugs could enhance the efficacy of therapeutic regimes. Moreover, curcumin is quickly metabolized under physiological condition, so the encapsulation of curcumin in nanoparticles could contribute to retaining its bioactivity.

### 2.5. Tumor Growth Suppression Results in Xenograft Models 

In the study, the SD mice were implanted tumors via a xenografted assay and their volume tracked from day mice were drug-treated to day 14 post-operation (Figure 6). The MCF-7 cancer cells proliferation inside the mice were confirmed with a superoxide dismutase 2 (SOD2) assay which only responded to a human source in immunohistochemical staining, in which the cancer cell nuclei were expressed in pale violet.

Regarding the study, tumor-bearing immunodeficient rodents maintained thie stabilized volume and were injected with NaCl 0.1%, Hep-F127, CDDP and Hep-F127/CDDP and Hep-F127/CDDP/Cur. The tumor-bearing mice were examined by measuring its body-weight and tumor-growth. In the NaCl 0.1% and Hep-F127 controlled groups, the rodents peculiarly maintained the balance or slightly increased in weight as seen in Figure 7, while the body weight of the three drug-treated models decreased. It is worth noting that in the comparison among drug-delivered nano-gels, a significant difference of body weight loss was found in the CDDP-treated group at day 10 (m = 21.94 ± 4.08 mg) and was followed by the death of three of five mice. Moreover, their injection sites displayed significant inflammation that gradually led to necrosis after 10 days. These phenomena could indicate a high cytotoxicity or side effects of CDDP in the CDDP-treated immunodeficiency rodents.

Interestingly, tumor volume growth was strikingly suppressed in rodents administered CDDP, Hep-F127/CDDP and Hep-F127/CDDP/Cur. In contrast, a high increase of around 52.12 ± 4.27 mm^3^ and 65.78 ± 3.91 mm^3^, respectively, was observed in rodent tumours treated with NaCl 0.9% and Hep-F127 (Figure 8). In a separate series of drug-tests, after 48 h the tumor volume of Hep-F127/CDDP/Cur group speeded up the gradual decline and overtook than those of the CDDP and Hep-F127/CDDP groups, to reach a nearly 80% size decrease at day 14. 

The reduction range in the observation is a supplemental attribution along with Figure 7 to indicate that dual drug-delivery system was the most effective in inhibiting tumor growth among those abovementioned regimens. 

In the study, tissues resected from MCF-7 xenograft rodents were subjected to histological analysis to further examine the MCF-7 density in tumor tissue using histological H&E staining. The results showed that in the control (Figure 9a,b), the cancer cell density was very high after 14 days of experiments. Cisplatin injection treatment was stopped after 10 days due to the multiple mice deaths, so the cellular density wasn’t evaluated by tissue staining histology. Hep-F127/CDDP treatment showed a decrement in MCF-7 cancer cell density after 14 days, as seen in Figure 9c. For the dual drug Hep-F127/CDDP/Cur delivery system (Figure 9d), the general characteristics of the tissue structure indicated a very small incidence of cancer cells. The result could indicate that the combination with nano-curcumin in the CDDP-complex nanoparticle minimized the systemic toxicity compared to that of free CDDP and enhanced the cancer cell killing efficiency in the xenografted tumors. In addition, the injection sites of the dual drug-treated mice didn’t show necrosis after 14 days. These firm evidences verify the effectiveness of the dual drug delivery system incorporating nano-curcumin and hydrated cisplatin species against MCF-7 cancer cell growth in vitro and in vivo xenograft tumor models.

## 3. Materials and Methods

### 3.1. Materials

Pluronic F127 (12,600 MW), cisplatin (300.05 MW, 99.999%) and curcumin were obtained from Sigma-Aldrich (St. Louis, MO, USA.) Heparin sodium (12,000 MW), silver nitrate (AgNO_3_, 169.87 MW), 3-amino-1-propanol (Ami, 75.77 MW, D = 0.982 g/mL), 1,4-diaminobutane (DAB, 88.15 MW, D = 0.877g/mL), 4-nitrophenyl chloroformate (NPC, 201.56 MW), 1-ethyl-3-dimethylaminopropyl carbodiimide (EDC, 191.7 MW), N-Hydroxysuccinimide (NHS) were obtained from Acros Organics (Antwerpen, Belgium). 

### 3.2. Preparation and Characterizations of Copolymer

Thermo-sensitive Hep-F127 was prepared based on the process of Tran et al. [23]. Heparin-conjugated F127 copolymer was prepared via four intermediate reactions. First, Pluronic F127 was activated by NPC and then monosubstituted by DAB to obtain NPC-F127-OH. Secondly, the obtained copolymer was reacted with an aminated heparin to produce Hep-F127. The structure of the grafted copolymer was confirmed by NMR spectroscopy and its thermoresponsive properties were determined by Krafft temperature and CMC points with fluorescence intensity ratios of pyrene excitation bands (I1 nm/I3 nm as 334 nm to 337 nm) as a function of the concentration of Hep-F127 in aqueous solutions. CMC values were determined as the point of intersection between two linear lines of regression generated by a piecewise fitting function.

### 3.3. Dual Drug Encapsulation

Curcumin was dissolved in a solvent mixture of ethanol:dichloromethane (ratio of 7:3 *v/v*) under stirring while water was the medium used for Hep-F127 copolymer dissolution. After that, the curcumin solution was added dropwise into Hthe ep-F127 mixture (ratio 1:3 of *w/w*) with the help of ultra-sonication below its LCST. Then, the Cur-dispersed in Hep-F127 solution was evaporated and freeze-dried to attain the Cur-entrapped-copolymer. The Cur-loaded Hep-F127 has dispersed again in deionized water (DI) at 25 °C and lyophilized to keep the well-maintained nanostructure. Thereafter, the nano-Cur was dialyzed with a membrane (MWCO 3.5 kDa) at 37 °C against deionized water in triplicate.

Cisplatin (CDDP) was hydrolyzed following Gordon’s method [27] in which, the molecular ratio of AgNO_3_ was doubled that of CDDP. The reaction mixture was magnetically stirred in a dark room for 24 h under N_2_. The amount of added CDDP was calculated responding to moieties of the remaining anionic groups on Hep-F127. Then, precipitated nitrate silver was completely removed by centrifugation to obtain the hydrated CDDP. The hydrated CDDP was added dropwise into the nano Cur-encapsulated Hep-F127 solution and kept in a dark under nitrogen and magnetically stirred below 20 °C for 24 h. Afterward, the Hep-F127/CDDP/Cur was dialyzed with regenerated cellulose MWCO 3500–5000 Da dialysis bags against DI water at 37 °C and lyophilized to obtain the dual drug-embedded Hep-F127. The amount of conjugated CDDP in the thermosensitive polymer was calculated as approximately 42.5 ± 0.62% (ICP-AES) and the nanocurcumin loading efficiency was 70 ± 0.56% (HPLC). The formation of the nano-complexes was confirmed by TEM, zeta potential, and DLS. 

### 3.4. In-Vitro Release of Dual Drugs from the Thermo-Responsive Platform

The dual drug-encapsulated Hep-F127 was dispersed in DI water and then added to a 3500 Da dialysis bag and dialyzed in 12 mL of phosphate buffer saline (PBS 0.01M) solutions with pH 7.4 and pH 5.5 at 37 °C to investigate the cumulative drug release. At a predetermined time interval, 1 mL of dialyzed solution was withdrawn to determine the conjugated CDDP and entrapped Cur and replaced with an equivalent volume of PBS in each tube. The amounts of released CDDP and curcumin were calculated by ICP-AES and HPLC measurements, respectively.

### 3.5. Evaluation of the Antiproliferative Activity of Dual Drugs Delivery System

Hep-F127, free CDDP, free Cur, Cur-loaded Hep-F127, CDDP-complexed Hep-F127 and dual drug-encapsulated thermosensitive nanostructure were screened to test their cell growth inhibition capability. For IC_50_ evaluation, cell viability was analyzed under different material concentrations using a Sulforhodamine B (SRB) colorimetric assay [28]. The cytotoxicity assay was performed three times and the average value of the three measurements was taken.

### 3.6. In-Vivo Xenograft Tumor Models for Drug Testing

The experiments were conducted with Mus musculus var. albino mice weighing around 35–37 g which were supplied by The Institute of Drug Quality Control (Ho Chi Minh City, Vietnam). Immunodeficient mouse models were created and cared for by the Laboratory of Physiology and Animal Biotechnology and Laboratory of Tissue Engineering and Biomedical Materials (University of Science, VNU, Hochiminh City, Vietnam). The animal experiment protocol was approved by the Animal Care and Use Committee of the University of Science, VNU, Vietnam. MCF-7 cells (107) in 100 µL NaCl were injected subcutaneously on one back side of each mouse and the tumor size of each mouse was measured 5 days after cell transplantation. Then, at a stably tumor size-measured stage, mice were randomly divided into five groups with five mice per group: control, CDDP, Hep-F127, Hep-F127/CDDP, Hep-F127/CDDP/Cur treated mice. NaCl (100 µL) was injected intravenously into the control mice while 3 mg of CDDP/kg, 12 mg Hep-F127, 15 mg of Hep-F127/CDDP (with around 3 mg of CDDP in the nanocomplex) and 18 mg of Hep-F127/CDDP/Cur (with around 3 mg of CDDP and 3 mg nanocurcumin in the nanocarrier) were administed to four treated mice, respectively, per 5 days. The tumor size of the control and drug-treated mice were measured by calipers at the time of administrating the drug aside from every 2 days. Moreover, the tumor volume of both the drug untreated and treated groups was calculated based on the formula of Tomayko, in which the minor axis and the major axis of tumor were the required measurements. Additionally, weight was estimated at the similar time as the tumor measurements [29]. All mice were housed in clean cages and maintained according to the institutional guidelines on animal welfare. Mice with tumors were euthanized after 15 days, and the xenografted and materials-treated tumors were evaluated by superoxide dismutase 2 (SOD2) assay, immunohistochemical staining and hematoxylin & eosin (H&E) staining. Hematoxylin-stained cellular nucleic acids show a deep blue-purple color via a completely non-understood complexation. Nuclei are stained blue in eosin and the extracellular matrix is stained pink [30].

### 3.7. Characterizations

Fourier transform infrared (FT-IR) spectrophotometry (Nicolet 5700, Thermo Electron Corporation, Waltham, MA, USA) was carried out to determine the structure of the synthesized copolymer. The morphology of nanoparticles was investigated by TEM (JEM-1400, JEOL, Eching, Germany) at an accelerating voltage of 100 keV, equipped with an AMT XR40 digital camera with 2 × 2 K pixels. The colloidal solutions were dispersed in water at 25 °C, dropped onto a carbon-coated Cu grid (EM Sciences, Gibbstown, NJ, USA) and dried at 37 °C to observe morphology of the nanogels without staining. Dynamic light scattering (DLS) and zeta potential were measured by a SZ-100 nanopartica series instruments (Horiba, Kyoto, Japan) at 1000 ppm of copolymer solution. Drug loading efficiency and drug release behavior of the nanocarriers were evaluated by HPLC and inductively coupled plasma atomic emission spectroscopy (ICP-AES) following the Association of Analytical Communities (AOAC)−990.08 method. Cytotoxicity assays towards the human breast cancer cell line MCF-7 using a Sulforhodamine B (SRB) colorimetric assay were used to form formazan in incubated solutions and then the optical density of the solution was measured to evaluate the cytotoxicity.

## 4. Conclusions

A thermo-responsive Hep-F127 copolymer was synthesized to complex/encapsulate CDDP-OH and nano-curcumin to form a dual drug delivery system. Not only the efficiency of drug loading but also the efficacy of the slow controlled-release system were significantly improved due to the mutual interaction of hydrated CDDP and encapsulated nano-curcumin with the characteristics of the amphiphilic Hep-F127 copolymer. The anti-proliferative ability determined through in-vitro and xenograft tumor triala suggests further studies on the use of this delivery system against various cancer types as well as resistant cancers due to the synergistic activity of a bioactive phytochemical and anticancer drugs in the nanocarriers.

## Figures and Tables

**Figure 1 molecules-23-03347-f001:**
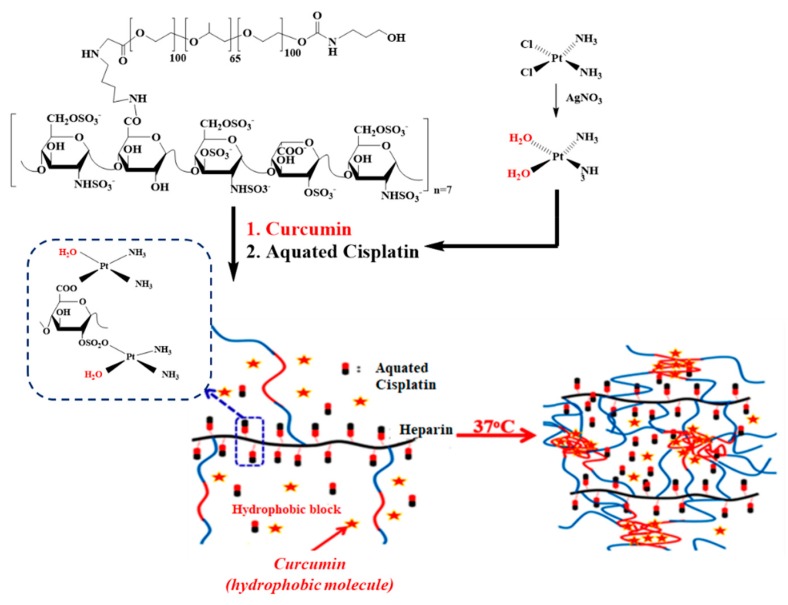
Thermosensitive dual drugs delivery nano-platform Hep-F127/CDDP/Cur (with CDDP being aquated cisplatin).

**Figure 2 molecules-23-03347-f002:**
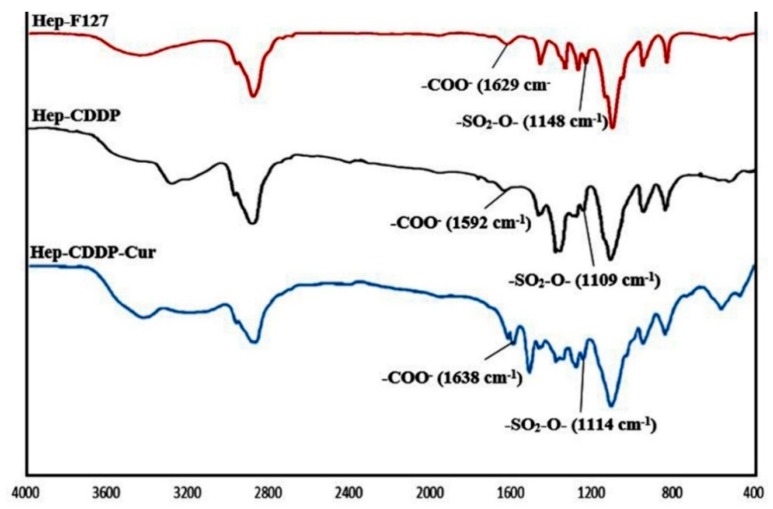
FT-IR spectra of Hep-F127; Hep-F127/CDDP; Hep-F127/CDDP/Cur.

**Figure 3 molecules-23-03347-f003:**
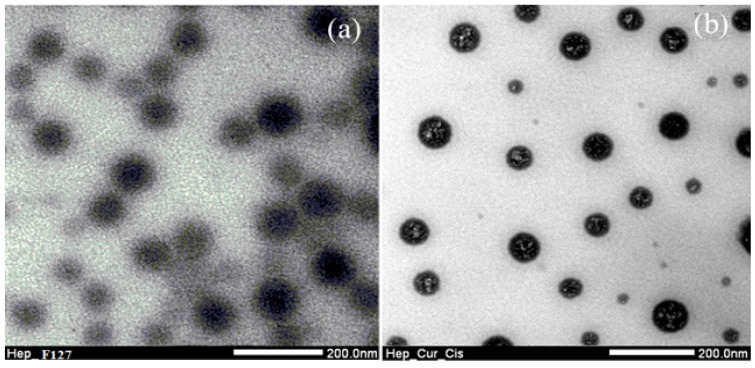
TEM images of the Hep-F127 copolymers (**a**) and dual drugs-loaded Hep-F127/CDDP/Cur nanocarriers (**b**).

**Figure 4 molecules-23-03347-f004:**
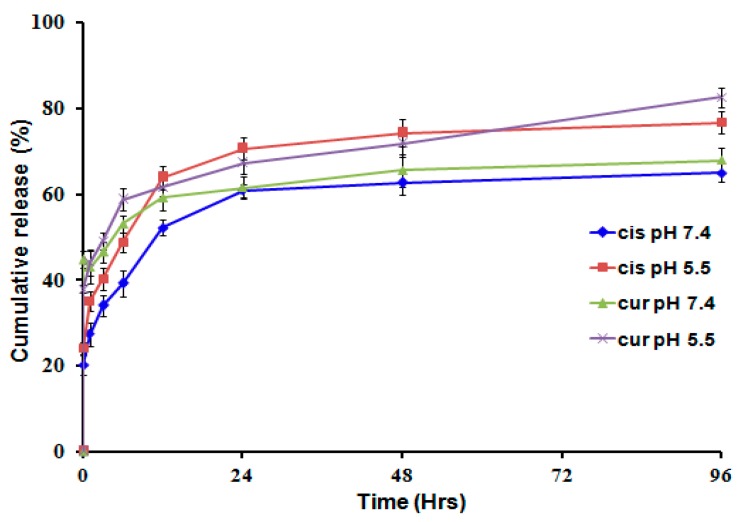
Release profiles of bioactive components from the dual drugs delivery system at pH = 7.4 and pH = 5.5.

**Figure 5 molecules-23-03347-f005:**
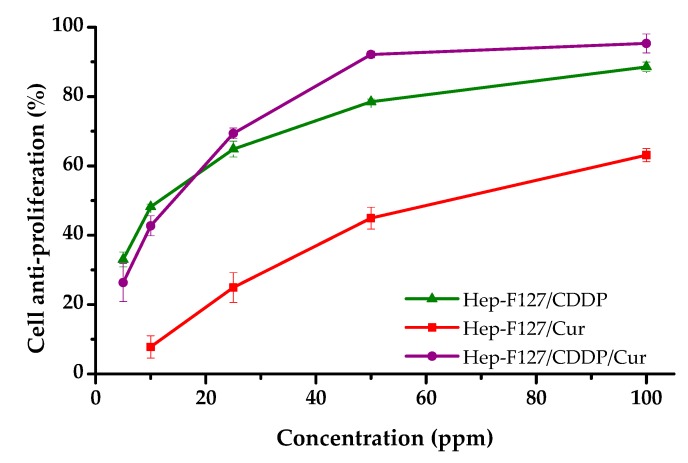
Inhibition of MCF-7 cancer cell growth of the drug delivery systems versus bioactive Cur and/or CDDP concentrations.

**Figure 6 molecules-23-03347-f006:**
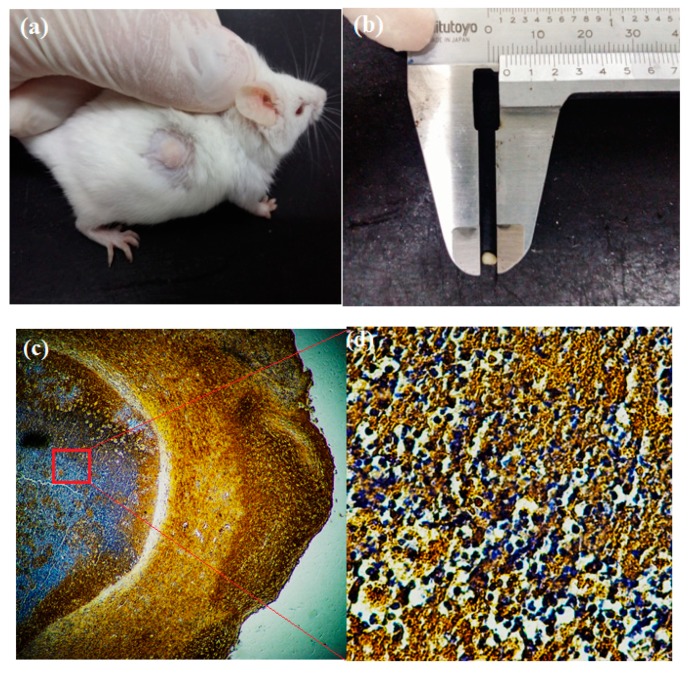
Tumor implanted by xenografted assay and Its characterized method: Implanted tumor before materials treatment (**a**), drug delivery system-treated tumor (**b**), SOD2 immunohistochemical staining of tumor (**c**,**d**).

**Figure 7 molecules-23-03347-f007:**
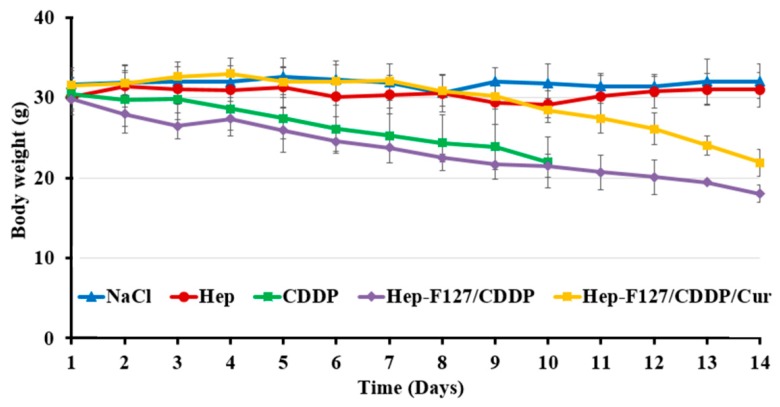
Body weight of the tumor-bearing immunodeficiency mice after materials-treated 14 days.

**Figure 8 molecules-23-03347-f008:**
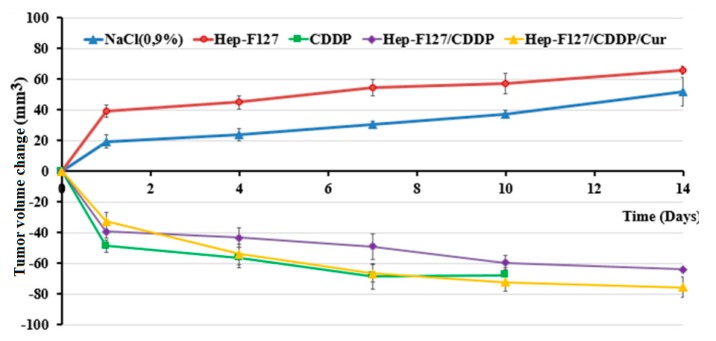
Volume behavior of tumors treated with normal saline control, CDDP, Hep-F127, Hep-F127/Cur, Hep-F127/CDDP and Hep F127/CDDP/Cur over time.

**Figure 9 molecules-23-03347-f009:**
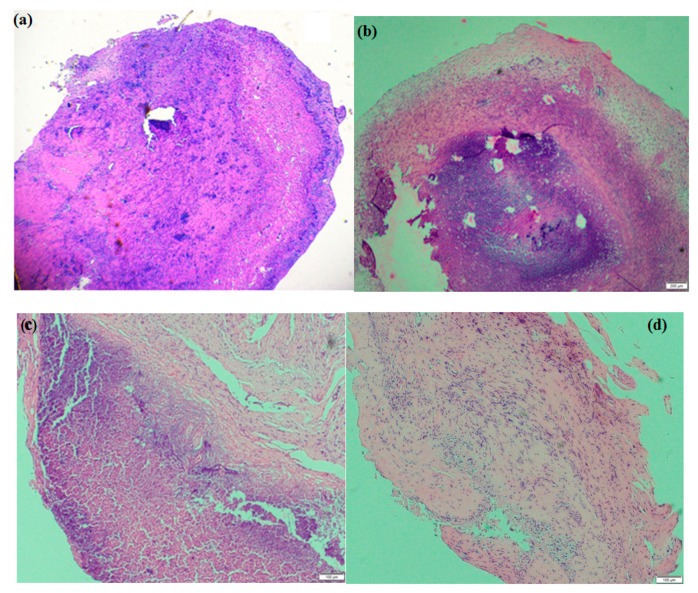
Histological tissue H&E analysis of after treatment: Hep-F127 (**a**), NaCl (**b**), Hep-F127/CDDP (**c**) and Hep-F127/CDDP/Cur (**d**).

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
