# Peer review of "Synergic Activity Against MCF-7 Breast Cancer Cell Growth of Nanocurcumin-Encapsulated and Cisplatin-Complexed Nanogels"

_molecules, 2018, doi:10.3390/molecules23123347_

Round 1

Reviewer 1 Report

This manuscript by Tran and coworkers describes investigation of cytotoxicity assay and of suppression of tumor growth by using Hep-F127/Cur/CDDP. The authors showed that this system was superior in comparison with Hep-Fq127/CDDP. Therefore, I recommend that this manuscript should be published in ‘‘molecules’’ after reconsideration of the following remarks. 1) Page3: The CMC of Hep-F127 was higher than that of F-127. Why this result is evidence of thermo-responsive behavior? The authors should mention clearly this point. 2) Page8: Why combination of nanocurcumin in the CDDP-complex nanoparticle can minimize the systemic toxicity? What is the role of curcumin (Cur) in this case? The authors should mention clearly this point. 

3) Page 6: “The SD” might be typo (“the SD”).

Author Response

Dear  - Replay to Reviewer 1

Thank you very much for processing our manuscript.

We have revised the manuscript according to comments from reviewers, answers reviewer’s opinions and queries are presented in the attached file below

Reviewer 2 Report

Report on article « synergic activity against MCF-7 breast cancer cell growth of nanocurcumin-encapsulated and cisplatin-complexed nanogels

This is an interesting study showing some promising results on encapsulate CDDP-And nanocurcumin (forming a dual drugs delivery system) in a Hep-F127 system, which can slowly release the drug and showed quite promising results on MCF-7 cancer cells, especially compared to the cisplatin drug, in term of side effects. This is an interesting study, which could be published, but after some revisions.

(1)  In my opinion, the authors should discuss more the drug release profile, in function of the polymer distribution. Indeed, what is the biodistribution of the polymers during the 24h first hours, since it is clear that the release of the drug will mainly occur during these? Is there accumulation at the tumor site? How long is the biopolymer eliminated? For me this point is really important, and has to be more deeply analyzed. Additionally, one other very important point: how is the drug release amount analyzed, I didn’t see this information anywhere, in the article (or I missed it?)? What is the method used?

(2) Figure 5: cytotoxicity assays: the measurement of cytotoxicity assays should at least be done three times (n=3), in triplicate. There is no information about it, but this is important, this should be done. Why are not the last point on the graph (100 pm concentration) for Hep-F127/cis and Hep-F127/cis/cur?

(3)  In addition to the body weight examination, do have the authors an idea if the polymer is toxic on healthy mice, at the concentration injected?

(4) Quality of figure 8 too low…The Figure 8 is not really clear. Why is the starting point “zero” at time 0? At the beginning of the injection of the compounds, the tumor volume is not zero, therefore this graph should be changed. Maybe it would be clearer to show the tumor evolution, before and after injection of the different compounds?

Author Response

Dear  - Replay to Reviewer 2

Thank you very much for processing our manuscript.

We have revised the manuscript according to comments from reviewers, answers reviewer’s opinions and queries are presented in the attached file below

Reviewer 3 Report

This manuscript Nguyen et al. demonstrates thermosensitive nanogels were shown synergistic anticancer activity against breast cancer. The author should carefully check the typographical and silly mistakes throughout the manuscript. Most importantly, the author generates poor quality of images without proper figure legends and description. Results section must be improved and results should be presented accurately. I suggest, the author should rewrite and revise the whole manuscript including methods and figures. Here are some comments to improve the quality of the manuscript;

1.     Related with cytotoxicity assay, please explain the figure legends in figure-5. The results indicated the concentration of drug was used as 100 µg/ml, however, the legends mentioned as PPM.

2.     The quality of the pictures at Figure 7 & 8 are very poor and need to replace. Did the tumor volume calculated in nm3 range as indicated in this line, “In contrast, a high-volume increment trend was observed in rodents’ tumour treated with NaCl 0.9% and Hep-F127 around 52.12± 4.27 nm3 and 65.78 ± 3.91 nm3, respectively (Fig. 8)”?

3.     What was the drug loading capacity in nanoparticles?

4.     Minor points:

a.     Some acronyms and abbreviations in the text need to be defined the first time that are mentioned; NCI, PLGA, DOX, Cur, CDDP, Hep-F127, LCST etc.

b.     The author should use either uppercase or lowercase in drug name, i.e. “Of all phytocompounds, curcumin……. consideration” and in another sentence “Due to the prominence ……. Curcumin and Cisplatin were investigated in this article”

c.     Introduction line 3, please check that statement is right. It is well-known that Erlotinib and Gefitinib are generally indicated as first line treatment in lung cancer.

d.     Correct the uppercase; Hepatocellular carcinoma > hepatocellular carcinoma

e.     Please correct section 3.2; Tran and et al [23] > Tran et al [23] (Assuming “AND” is not part of name)

f.      Please correct section 3.3; After that, the curcumin solution was drop-wise > After that, the curcumin solution was added drop-wise.

g.     Free-dried or freeze dried?

h.     Mention the pore size of membrane that used in section 3.3.

i.      Please correct this sentence as MCF-7 is not a lung cancer cell line: “Cytotoxicity assay towards human lung cancer cell line MCF-7 using Sulforhodamine B……”

Author Response

Dear  - Replay to Reviewer 3

Thank you very much for processing our manuscript.

We have revised the manuscript according to comments from reviewers, answers reviewer’s opinions and queries are presented in the attached file below

Reviewer 4 Report

The paper “Synergic activity against MCF-7 breast cancer cell growth of nanocurcumin-encapsulated and cisplatin-complexed nanogels” from Ngoc The Nguyen et al., focuses on the co-deliver of cisplatinum and curcumin included into nanogels based on pluronic and heparin.

The topic of this paper is interesting but the whole text and some of the performed experiments do not reach an acceptable qualitative standard for the scientific community. I suggest to reject the paper. I also suggest to the Authors to perform a really deep revision of the document and to critically discuss the gained results also strongly improving the experimental part of the paper.

Please find below some suggestion to improve Your paper.

The whole text should be revised by an English mother language.

CDDP has to be defined as Cisplatin where first appears.

“aqated cisplatin” in figure 1 should be defined in the caption.

Line 218: the synthesis of the co-polymer should be briefly described. Furthermore, scheme 1 is not present in the paper and, even if it would be referred to Figure 1, it does not describe the co-polymer synthesis.

Line 220: the meaning of the sentence “…was evaluated via point…” is not clear.

Line 228: “sonication in the range of LCST”, above or below? Please explain or , better, indicate the setted temperate for drug encapsulation.

Line 229: “was vapor rotated and free-dried” please revise this sentence fixing both “vapor-rotated” and “free dried”.

Line 230: “continuously lyophilized”?  “to maintain the stability of nanostructure”? please fix.

Line 239: “Afterward, the final product was similarly under the dialysis by the regenerated cellulose”, not clear.

Line 242: “was approximately calculated around 42.5 % (ICP-AES) and nanocurcumin-loaded efficiency was around 70%” this is not scientific. Please indicate the percentage by including the SD and avoiding “approximately calculated around”.

Line 245: “was dispersed in solution”, which kind of solution?

Line 246: “kinetic drug release” did You calculate the release kinetic or the cumulative release?

Line 246: “by dialysis in phosphate buffer saline”, please indicate the volume of PBS used. How did You assure sink conditions without using a surfactant?

Line 270: “the volume was calculated” the volume of what?

Line 281: please add the procedure for TEM sample preparation (staining, dispersion concentration, drying of the sample and so on).

Line 282: please include the procedure for DLs and zeta potential evaluation.

Line 283: please include UV-VIS conditions. In particular accounting on the determination of the two drugs, calibration curves, wavelength(s) and so on.

Line 290: “was synthesized with the facilitation”, please revise.

Line 292: “but also controllably slow-release system”. Please rivise.

Line 82: “the hydrophilic CS and PEO segments” what is CS?

Line 83: “provided the grafted copolymer an opportunity to self-assemble in aqueous solutions”, please rivise.

Line 89: “decreases with the increase of Hep-F127 89 concentration at 37°C”, please indicate why 37 °C linking it to LCST.

Line 90 and whole text: it would be better indicating the CMC as CAC.

Line 92: “The reason may be that the Hep chain is negatively charged and the electrostatic repulsion results in the micellization of Hep-F127 at higher concentration.” This explanation is not supported by literature and, to be honest, is also far from my opinion. In fact, by increasing the hydrophilic portion of a surfactant-like molecule its theoretic HLB should increase determining an increase in the hydrophilicity of the polymer that would be more prone to stay in “solution” so lowering its tendency in hydrophobic assembling. On the other side, by increasing the hydrophilic behaviors of the polymer it would also bring to stronger hydrophobic interactions of the PPO portions. Obviously, these two assumptions are contrasting. Another factor that could influence a lower hydrophobic interaction of the co-polymer is the increased steric hindrance between the (bigger) polymer chains. Please check the literature for similar systems.

Line 97: “To strongly confirm the successful structure of nano-complex” please revise.

Line 121: “hydrophobic and negative-charged properties of curcumin” please explain why CUR should be considered negative-charged.

Line 128-130: please completely rewrite this sentence.

Paragraph 2.3: the release studies should be fully revised also including the dissolution curves of the drugs in the release medium. Furthermore, is not clear how the study was performed and how the drugs were detected. I’m sorry to say that is part is not scientifically performed.

Author Response

Dear  - Replay to Reviewer 4

Thank you very much for processing our manuscript.

We have revised the manuscript according to comments from reviewers, answers reviewer’s opinions and queries are presented in the attached file below

Round 2

Reviewer 2 Report

The authors took all the comments and corrections very seriously, and made a really good job on answering all the comments and doing all the required corrections. I am now happy with the quality of the manuscript, and recommend it for publication. 

Reviewer 3 Report

Since the authors addressed most of the questions from the reviewers and corrected accordingly, I would like to accept this manuscript.

Reviewer 4 Report

Thank You to the Authors for accepting some of the suggestions I provided. There are still different points not fixed and that i consider fundamental. Among those, sink conditions not reported and talking about "colloidal curcumin" in its free form is really hard to read and a number of other points that i still evidenced in my long report.

I would not accept this paper even after the revision. The Editor will point better than me on this decision.